# Determinants of COVID-19-Related Length of Hospital Stays and Long COVID in Ghana: A Cross-Sectional Analysis

**DOI:** 10.3390/ijerph19010527

**Published:** 2022-01-04

**Authors:** Shirley Crankson, Subhash Pokhrel, Nana Kwame Anokye

**Affiliations:** Division of Global Public Health, Department of Health Sciences, College of Health, Medicine and Life Sciences, Brunel University London, Uxbridge, London UB8 3PH, UK; Shirley.Crankson@brunel.ac.uk (S.C.); Subhash.Pokhrel@brunel.ac.uk (S.P.)

**Keywords:** long COVID, COVID-19, hospitalisation, determinants, Ghana

## Abstract

Objectives: There is paucity of data on determinants of length of COVID-19 admissions and long COVID, an emerging long-term sequel of COVID-19, in Ghana. Therefore, this study identified these determinants and discussed their policy implications. Method: Data of 2334 patients seen at the main COVID-19 treatment centre in Ghana were analysed in this study. Their characteristics, such as age, education level and comorbidities, were examined as explanatory variables. The dependent variables were length of COVID-19 hospitalisations and long COVID. Negative binomial and binary logistic regressions were fitted to investigate the determinants. Result: The regression analyses showed that, on average, COVID-19 patients with hypertension and diabetes mellitus spent almost 2 days longer in hospital (*p* = 0.00, 95% CI = 1.42–2.33) and had 4 times the odds of long COVID (95% CI = 1.61–10.85, *p* = 0.003) compared to those with no comorbidities. In addition, the odds of long COVID decreased with increasing patient’s education level (primary OR = 0.73, *p* = 0.02; secondary/vocational OR = 0.26, *p* = 0.02; tertiary education OR = 0.23, *p* = 0.12). Conclusion: The presence of hypertension and diabetes mellitus determined both length of hospitalisation and long COVID among patients with COVID-19 in Ghana. COVID-19 prevention and management policies should therefore consider these factors.

## 1. Introduction

The COVID-19 pandemic continues to unfold, causing significant morbidity and mortality globally. As of 10 October 2021, more than 236 million cases and over 4 million deaths had been confirmed worldwide [1]. Although the number of weekly reported cases and deaths has declined considerably over the past few weeks [2], there is still a need for research-informed effective interventions to curtail the outbreak altogether. The literature is currently inundated with several COVID-19 studies to address the COVID-19 menace; however, most of these studies focus on the risk of COVID-19 infections and COVID-19-related mortality [3,4,5,6]. Thus, there is limited empirical research on other equally important COVID-19-related outcomes, such as COVID-19 length of hospital stay (LOS) and long COVID.

COVID-19-related LOS has a mammoth impact on global health systems, including overburdened health resources, unprecedented demand for health professionals, workers burnout and increased health financing and medical cost [7]. For instance, in Ghana, the average medical cost of managing a single case of COVID-19, based on the national COVID-19 treatment protocol, was estimated at 11,925 USD (United States dollar) as of October 2021 [8]. For a patient with critical COVID-19 disease that required specialised hospital care, this cost was estimated to increase by at least two folds during the period of care [8]. Moreover, by extension, this cost could further increase with prolonged LOS. Although Ghana’s government absorbs the medical cost of COVID-19 case management to reduce the economic burden of COVID-19 on its citizens, potential productivity loss due to prolonged COVID-19 LOS could still have dire economic consequences for individuals and organisations, as evident in other jurisdictions [9]. Apart from these financial burdens, COVID-19 LOS also poses a significant risk for hospital-acquired infections, further spiralling the overall COVID-19-related health burden [10].

Regardless of this enormous burden imposed by COVID-19 LOS on health systems and individuals, only a few studies have contributed evidence to address COVID-19-related LOS [7,11,12,13,14]. Nonetheless, two of these studies mainly summarised COVID-19 LOS distribution among the study participants [7,11]. In addition, while the others further explored the determinants of prolonged COVID-19 LOS, they involved a relatively smaller size, limiting the generalisation of findings [12,13,14]. Furthermore, apart from the limited scope described above, the studies were also conducted mainly in high-income countries (HIC). These limitations create two main gaps that require urgent attention. First, there is a need for research involving a larger sample size on determinants/factors associated with prolonged COVID-19 to address this paucity and provide a research balance on COVID-19, particularly as the disease continues to pose a significant burden on health systems globally. Second, evidence on determinants of COVID-19-related LOS from low- and middle-income countries (LMICs) is required for critical contextual policies, particularly given the resource constraints of most health services in LMICs, and the influence of geographic context on research acceptability and implementation [15].

Another significant burden associated with the COVID-19 outbreak is its emerging long-term sequelae, long COVID. Long COVID is defined as persistent COVID-19 symptoms lasting more than four weeks after getting COVID-19 and not explained by any alternative diagnosis [16]. It is further categorised into two groups: 1. ongoing symptomatic COVID-19 symptoms lasting 4–12 weeks and 2. post COVID-19 syndrome symptoms lasting more than 12 weeks [16,17]. Symptoms of long COVID include fatigue, cough, breathlessness and body aches [18]. Long COVID tends to pose an additional medical, economic and psychological burden on individuals and health systems, with the most deprived individuals and institutions likely to experience the worse impact [19]. For most deprived health systems that have yet to overcome the enormous implications of the COVID-19 outbreak, the added burden of long COVID could cripple service delivery considerably, further worsening health inequalities. Moreover, potential prolonged long COVID-related economic productivity loss could present an existential threat to business. As long COVID is anticipated to persist, especially as the pandemic is still ongoing, it is crucial to identify its determinants to inform related mitigating interventions.

Therefore, this study aimed to contribute to the limited arsenal of knowledge on COVID-19-related LOS and long COVID by investigating its determinants in Ghana. Understanding these phenomena could help flag potential risk factors for both LOS and long COVID and provide policy menus to address them. For instance, the findings on LOS could provide data for forecasting demands for specialist care and hospital resources, especially hospital beds demand, one of the challenges most government hospitals face in Ghana. Finally, this study’s findings could also potentially help alleviate the overall disease burden of long COVID and COVID-19 LOS on individuals and health systems in Ghana.

## 2. Methods

### 2.1. Data Source

Data for this study were accessed from Ga East Municipal Hospital (GEMH), the main COVID-19 treatment centre in Ghana. The centre obtains data from its patients as part of clinical practice. This data includes the patient’s age, sex, employment status, clinical symptoms, medical history, admissions dates, and treatment outcomes, i.e., discharges, transfers, and deaths and follow-ups. At the time of accessing the GEMH dataset, the centre had seen over two thousand patients with PCR-confirmed COVID-19 diagnoses. For this study, the data of admitted COVID-19 patients in GEMH who were transferred to other hospitals for further admission and treatment were excluded from the analysis because it was difficult to estimate their total COVID-19 LOS. Therefore, the data of 2334 patients seen at the centre from March 2020 to August 2021 were included in this analysis.

### 2.2. Variables

#### 2.2.1. Dependent Variables

The dependent variables in this study were COVID-19 LOS and long COVID. COVID-19 LOS was assessed as the number of days the patient stayed in the hospital due to COVID-19. Thus, it was explored as a count variable. Long COVID was operationalised as patients who still reported symptoms of COVID-19 4 weeks after the initial illness with no alternative medical diagnosis [16,17]. Hence, it was examined as a binary outcome. Those without long COVID were coded “0” and those with long COVID “1”.

#### 2.2.2. Independent Variables

The independent variables included age, sex, nationality, marital status, education level, employment status and comorbidities, explanatory variables consistent with the health determinants espoused in the Dahlgren and Whitehead (1993) determinants of health framework. The patients’ ages were described as the number of years lived and were categorised into three groups: 0–29 years, 30–59 years and ≥60 years. Their sex was described as either “men” or “women”, and their nationality was their country of origin. This study was specified as “Ghanaian” for those originating from Ghana and non-Ghanaian for those with different nationalities. The marital status described whether the patient was single, married, cohabiting, divorced/separated or widowed. In addition, the employment and education level described the employment status of the patient and their educational attainment, respectively. Finally, comorbidity was defined as any other existing clinical condition of the patient. It included mainly hypertension (HPT), diabetes mellitus (DM), HPT/DM (hypertension and diabetes mellitus), pulmonary diseases (asthma, chronic obstructive pulmonary diseases (COPD), tuberculosis and pneumonia), neoplasms (all benign and malignant tumours), gastrointestinal diseases (peptic ulcer diseases, diarrhoea, lactose intolerance and gastroesophageal reflux disease), cardiovascular diseases (coronary heart disease and peripheral artery disease) and neurological diseases (stroke and seizures).

### 2.3. Data Analysis

The data were analysed in three main stages: (1) descriptive analysis; (2) bivariate/hypothesis testing; (3) regression analysis. The descriptive analysis examined the accuracy of the data and summarised the characteristics of the variables. The normality of the count variable was checked with the Kolmogorov–Smirnov and Shapiro–Wilk normality tests and described with either means and standard deviations (SD) or median and interquartile range (IQR). For the categorical variables, proportions were used to describe their characteristics. Subsequently, Mann–Whitney *U*-test, Kruskal–Wallis and chi-square tests were conducted to examine associations between the independent and dependent variables in bivariate analyses.

For the regression analysis, generalised linear models with a negative binomial with log link, robust to skewed count outcomes with a variance significantly higher than the mean, were fitted to identify determinants of COVID-19-related LOS. Additionally, binary logistic regression was conducted to examine determinants of long COVID. Finally, these models were again fitted in further analysis to compare determinants of long COVID and COVID-19-related LOS among men and women to inform gender/sex-sensitive interventions. Before fitting the model, the Hosmer–Lemeshow test was conducted to ensure a good model fit. Moreover, all assumptions for the two models were checked and confirmed before fitting the models. The data analysis was performed with SPSS version 26, and the statistical significance level was set at *p* < 0.05.

## 3. Results

### 3.1. Descriptive Characteristics and Bivariate Findings

Of the 2334 patients, 60.1% were men and 39.9% were women. The majority were aged from 30 to 59 years (57.5%), married (55.9%), employed (73.6%) and were tertiary educated (46.5%). In addition, most of them were Ghanaians (74.3%). Among the non-Ghanaians, Nigerians (*n* = 152) constituted the majority, followed by Americans (*n* = 57) and British (*n* = 49). Most of the patients had no comorbidities (81.5%), and among those with comorbidities, HPT (10.2%) and HPT/DM (3.4%) were the most common. Many of the patients survived from COVID-19 (98%). Amongst the nearly 2% (*n* = 45) that died, 46.7% (*n* = 21) had HPT, 24.4% (*n* = 11) had DM and 15.6% (*n* = 7) had both, HPT/DM.

For COVID-19-related LOS, the maximum length of hospital stay was 74 days, and the average LOS of the total sample was 4.73 ± 5.93 days. Those who died from COVID-19 spent fewer days (4.53 ± 4.19 days) in hospital than those who survived (4.73 ± 5.96 days). Among the survivors, the patients with comorbidities spent three days longer in the hospital compared to those with no comorbidities. However, among those who died, the patients with comorbidities spent three days less in hospital compared to those with no comorbidities. Similarly, among the male patients that died, those with comorbidities spent about two days less in hospital compared to those with no comorbidities.

Furthermore, about 2% (*n* = 50) of the patients had long COVID. Of this, many were 30–59 years old (*n* = 30), employed (*n* = 37) and had no comorbidity (*n* = 34). Comparatively, the men were diagnosed with long COVID 2.6% of the time whereas the women were diagnosed 1.5% of the time. Moreover, while 8.7% of the patients with hypertension and diabetes were diagnosed with long COVID, none of the patients with neoplasms, gastrointestinal, pulmonary and cardiovascular diseases were diagnosed with long COVID. In the bivariate analysis, age, nationality, education level and comorbidities were all statistically associated with long COVID. See Table 1 and Table 2 and Figure 1 for the patients’ characteristics.

### 3.2. Determinants of COVID-19-Related LOS

In the negative binomial log link regression, COVID-19 patients with tertiary education were less likely to stay longer in hospital due to COVID-19 than were those with no formal education (B = 0.55, 95% CI = 0.39–0.77, *p* = 0.00). Similarly, COVID-19 patients with HPT/DM had 1.82 times increased likelihood of prolonged COVID-19-related LOS compared to those with no comorbidities (95% CI = 1.42–2.33, *p* = 0.00). Additionally, those with HPT (B = 1.26, 95% CI = 1.08–1.47, *p* = 0.003), DM (B = 1.37, 95% CI = 0.99–1.88, *p* = 0.05) and gastrointestinal diseases (B = 2.08, 95% CI = 1.25–3.46, *p* = 0.005) were more likely to spend about a day longer in COVID-19-related hospitalisations compared to those with no comorbidities.

In the subgroup analysis, the risk of prolonged COVID-19-related LOS increased by a day in women aged 30–59 years old compared to those aged 0–29 years old (95% CI = 1.07–1.62, *p* = 0.009). Moreover, the men (B = 1.67, 95% CI = 1.16–2.41, *p* = 0.006) and women (B = 1.88, 95% CI = 1.34–2.63, *p* = 0.000) with HPT/DM spent nearly 2 days longer in hospital due to COVID-19 compared to those with no comorbidities. However, on average, men with HPT (B = 1.26, 95% CI = 1.02–1.54, *p* = 0.03), DM (B = 1.47, 95% CI = 0.99–2.17, *p* = 0.05) and gastrointestinal diseases (B = 2.82, 95% CI = 1.41 –5.61, *p* = 0.003) spent more days in COVID-19-related hospitalisations than did the women with these conditions. See Table 3 for the determinants of COVID-19-related LOS findings.

### 3.3. Determinants of Long COVID

The findings of the logistic regression showed that women are less likely to be diagnosed with long COVID than are men (OR = 0.52, 95% CI = 0.27–0.99, *p* = 0.05). Moreover, patients with primary (OR = 0.73, 95% CI = 0.01–0.66, *p* = 0.02), secondary/vocational (OR = 0.26, 95% CI = 0.09–0.77, *p* = 0.02) and tertiary education (OR = 0.23, 95% CI = 0.07–0.72, *p* = 0.12) had lower odds of long COVID diagnosis compared to those with no formal education. Further, for patients with HPT/DM, the odds of being diagnosed with long COVID were four times more than for those with no comorbidities (95% CI = 1.61–10.85, *p* = 0.003). The Hosmer–Lemeshow test for this model was 0.91, indicating a good model fit.

In extended analyses, men with secondary/vocational (OR = 0.23, 95% CI = 0.06–0.83, *p* = 0.02) and tertiary education (OR = 0.17, 95% CI = 0.04–0.69, *p* = 0.12) were statistically less likely to have long COVID diagnosis than were women with the same educational level. However, women with HPT/DM (OR = 5.69, 95% CI = 1.08–30.16, *p* = 0.04) had a marginally higher odds of long COVID diagnosis compared to men with HPT/DM (OR = 4.58, 95% CI = 1.32–15.93, *p* = 0.02). Finally, the Hosmer–Lemeshow test for the men (0.63) and women (0.71) subgroup analyses was more than 0.05, suggesting a good model fit. See Table 4 for the determinants of long COVID.

## 4. Discussion

This study aimed to explore the determinants of COVID-19-related LOS and long COVID in Ghana. The data of 2334 patients accessed from the electronic medical records of the main COVID-19 treatment centre in Ghana—GEMH—from March 2020 to August 2021 were included in the cross-sectional analyses. Of the 2334 COVID-19 patients, nearly 2% died, about 2% had long COVID, and their average COVID-19-related LOS was 4.73 ± 5.93 days. Averagely, the patients who died spent about 2 h less in hospital than did those who survived; however, their mean difference was insignificant (*p* = 0.24). The decreased LOS among the nonsurvivors is consistent with recent evidence [7]. Moreover, the patients with comorbidities who died spent three days less in hospital compared to those with no comorbidities who died, regardless of their sex. Similarly, the patients with comorbidities who died stayed three days less in the hospital than did those who survived. This finding is also consistent with current literature [11]. Furthermore, of the 2% of patients that had long COVID, the majority were men, married, employed, and had HPT/DM. Also, half of them had secondary/vocational training.

In the regression analysis, comorbidities were identified as determinants of both COVID-19 LOS and long COVID in the general population. Additionally, men with HPT, DM and gastrointestinal diseases were statistically more likely to stay longer in hospital than were women with similar conditions. In the logistic model, women had lower odds of long COVID diagnosis compared to men. This observation may be due to increased tendencies for the men to continue reporting persistent COVID-19 symptoms compared to the women. Nonetheless, this finding is yet to be corroborated in current research, probably due to the apparent paucity of literature on determinants of long COVID. The association between comorbidities, long COVID and COVID-19 LOS is consistent with several studies [11,12,20]. One possible explanation for these observed associations is the comorbidities’ prolonging effect on the overall COVID-19 disease experience, primarily due to related decreased innate immunity [21,22]. Moreover, the association between comorbidities and COVID-19 could be bidirectional as any of them could exacerbate the other and further prolong COVID-19 recovery [23].

One notable nuance in Ghana’s COVID-19 experience, evidenced in this study’s finding, is the relatively lower COVID-19-related mortality (1.9%, *n* = 45) and morbidity—from the LOS and long COVID findings. These observed lower mortality and morbidity rates are probably due to the age characteristics (mean age = 40.2 ± 15.9) of the patients included in this study, especially as ageing has been identified as a common determinant of COVID-19 morbidity and mortality [24,25]. Ghana, like most developing countries, has a smaller aged population (narrow top expansive age pyramid) than that of most developed countries, ostensibly due to a lower life expectancy associated with poor health structures and endemic diseases, such as malaria, in the former [26,27]. This rather generally smaller aged population in Ghana could have invariably influenced the proportion of the aged in the GEMH dataset, and it could have inherently influenced this study’s findings. Moreover, the GEMH dataset had fewer COVID-19 patients aged ≥60 years (*n* = 327) compared to those aged <60 years (*n* = 2007). Therefore, as already indicated, the lower mortalities and morbidities identified in this study are more likely a reflection of the younger population in Ghana and not necessarily due to a lower risk of COVID-19 infections and its subsequent mortality morbidity. To buttress this point further, many of the nonsurvivors in this study were sixty years old and above (See Figure 1).

Finally, this study’s findings on the significant associations between comorbidities and COVID-19-related LOS and long COVID have implications for health policies and planning in Ghana. For example, it indicates an urgent need for Ghana’s government to prioritise persons with comorbidities in its COVID-19 preventive strategies to reduce their overall risk of COVID-19. Similarly, specialised services, such as diabetic and hypertension clinics, must be implemented at national, metropolitan and district levels to give focused interventions to all persons with comorbidities to reduce their risk of COVID-19 prolonged morbidity and improve their overall life expectancies. Furthermore, this study’s findings could serve as a basis for resource mobilisation, planning and allocation to avoid potentially overwhelming health systems in Ghana due to prolonged COVID-19 LOS. More specifically, the findings could help provide contingency plans around bed occupancy in Ghana, especially as Ghana continues to face the “no bed syndrome” in most government hospitals. Moreover, effective health promotion interventions targeted at the high-risk groups identified in this study could help reduce the incidence of long COVID in Ghana to prevent another COVID-19-related epidemic.

### Study’s Strength and Limitation

To the best of the researchers’ knowledge, this is the first study exploring the determinants of COVID-19 LOS and long COVID in Ghana. They, therefore, offer new findings to inform current COVID-19 policy responses in Ghana. In addition, using a secondary dataset in the analyses ensured a higher sample size, thus increasing the statistical power and external validity of this study. Nonetheless, the findings of this study must be interpreted with caution due to the retrospective nature of the data. Moreover, the study focused on only one COVID-19 centre in Ghana, albeit it is the main treatment centre. Thus, the generalisations of the findings to all persons with COVID-19 in Ghana is restricted. Additionally, the study’s cross-sectional nature limited the researcher from drawing causal associations between the independent and dependent variables.

## 5. Conclusions

Given the paucity of research on the determinants of long COVID in the literature, this study’s findings provide critical evidence with considerable implications for further research on long COVID management. Moreover, the COVID-19-related LOS findings could inform health resource planning for most health services, particularly in Ghana and the wider LMICs with similar health characteristics to those of Ghana. Therefore, it is recommended that such studies be replicated in other LMICS to offer contextual findings and potential subsequent interventions. Moreover, future studies could explore the influence of factors such as smoking, diet and physical activity on COVID-19 LOS and long COVID to ensure comprehensive literature coverage.

## Figures and Tables

**Figure 1 ijerph-19-00527-f001:**
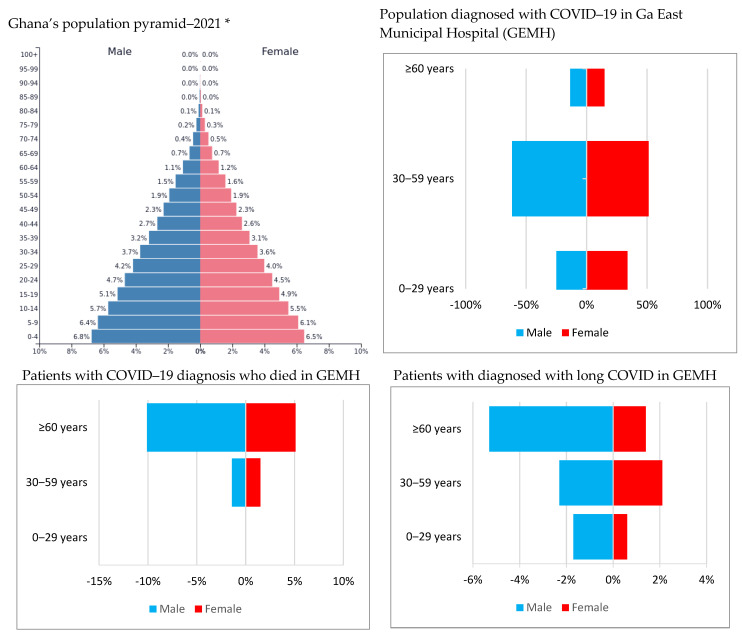
Population pyramids showing the burden of COVID-19 in Ghana by age and gender. (* Image source: https://www.populationpyramid.net/ghana/2021/, accessed on 30 December 2021).

**Table 1 ijerph-19-00527-t001:** Sample characteristics and bivariate findings (*N* = 2334).

Variables	Number (%)	Long COVID		
		No Number (%)	Yes Number (%)	*p*-Value
Age				<0.05
0–29 years	664 (28.4)	656 (98.8)	8 (1.2)	
30–59 years	1343 (57.5)	1313 (97.8)	30 (2.2)	
≥60 years	327 (14)	315 (96.3))	12 (3.7)	
Sex				0.08
Men	1402 (60.1)	1366 (97.4)	36 (2.6)	
Women	932 (39.9)	918 (98.5)	14 (1.5)	
Nationality				<0.05
Ghanaian	1733 (74.3)	1689 (97.5)	44 (2.5)	
Non-Ghanaian	601 (25.7)	595 (99)	6 (0.9)	
Marital status				0.32
Single	872 (37.4)	859 (98.5)	13 (1.5)	
Married	1305 (55.9)	1274 (97.6)	31 (2.4)	
Cohabiting	16 (0.7)	15 (93.8)	1 (6.2)	
Divorced/separated.	82 (3.5)	79 (96.3)	3 (3.7)	
Widowed	59 (2.5)	57 (96.6)	2 (3.4)	
Education level				<0.01
No formal	45 (1.9)	40 (88.9)	5 (11.1)	
Primary	156 (6.7)	155 (99.4)	1 (0.6)	
Secondary/vocational	1048 (44.9)	1023 (97.6)	25 (2.4)	
Tertiary	1085 (46.5)	1066 (98.2)	19 (1.8)	
Employment status				0.94
Employed	1717 (73.6)	1680 (97.8)	37 (2.2)	
Unemployed	617 (26.4)	604 (97.9)	13 (2.1)	
Comorbidities				<0.01
No comorbidity	1902 (81.5)	1868 (98.2)	34 (1.8)	
Hypertension (HPT)	239 (10.2)	233 (97.5)	6 (2.5)	
Diabetes mellitus (DM)	47 (2.0)	45 (95.7)	2 (4.3)	
HPT/DM (hypertension and diabetes mellitus)	80 (3.4)	73 (91.3)	7 (8.7)	
Neoplasms	4 (0.2)	4 (100)	0 (0)	
Neurological diseases	2 (0.1)	1 (50)	1 (50)	
Gastrointestinal diseases	17 (0.7)	17 (100)	0 (0)	
Pulmonary diseases	31 (1.3)	31 (100)	0 (0)	
Cardiovascular diseases	12 (0.5)	12 (100)	0 (0)	

**Table 2 ijerph-19-00527-t002:** Average LOS among COVID-19 survivors and nonsurvivors (*N* = 2334).

Groups	Number (*n*)	Survived (*N* = 2289)	Died (*N* = 45)	*p*-Value (LOS Mean Difference)
		LOS (Days) Mean (SD)	LOS (Days) Mean (SD)	
General population	2334	4.73 (5.96)	4.53 (4.19)	0.24
Patients with comorbidities	432	7.19 (6.45)	4.27 (4.09)	0.01
Patients with no comorbidities	1902	4.23 (5.7)	7.25 (4.86)	0.25
Female patients with comorbidities	197	7.25 (5.99)	4.15 (4.89)	0.06
Female patients with no comorbidities	735	4.09 (5.36)	9.00 (-)	0.26
Male patients with comorbidities	235	7.14 (6.85)	4.32 (3.77)	0.09
Male patients with no comorbidities	1167	4.31 (5.94)	6.67 (5.77)	4.97

Note: SD—standard deviation, (-)—no SD was calculated because that sample contained only one patient.

**Table 3 ijerph-19-00527-t003:** Determinants of COVID-19-related LOS (*N* = 2334).

Variables	General Population (*N* = 2334) Correlation Coefficient (B) (95% CI)	Men (*N* = 1402) B (95% CI)	Women (*N* = 932) B (95% CI)
0–29 years	(1,1)	(1,1)	(1,1)
30–59 years	1.18 (0.97–1.43)	0.934 (0.78–1.12)	1.32 ** (1.07–1.62)
≥60 years	1.079 (0.941–1.24)	1.15 (0.89–1.48)	1.27 (0.94–1.70)
Men	(1,1)		
Women	0.93 (0.85–1.02)		
Ghanaian	(1,1)	(1,1)	(1,1)
Non-Ghanaian	0.45 ** (0.41–0.51)	0.53 ** (0.46–0.61)	0.34 ** (0.29–0.41)
Single	(1,1)	(1,1)	(1,1)
Married	0.91 (0.80–1.03)	0.94 (0.8–1.11)	0.84 (0.69–1.02)
Cohabiting	1.441 (0.84–2.46)	1.88 (0.99–3.57)	0.69 (0.25–1.88)
Divorced/separated	1.20 (0.92–1.56)	1.24 (0.87–1.76)	1.09 (0.74–1.63)
Widowed	1.071 (0.79–1.46)	1.13 (0.72–1.79)	0.95 (0.62–1.46)
No formal education	(1,1)	(1,1)	(1,1)
Primary	0.61 (0.43–0.87)	0.63 (0.38–1.03)	0.604 (0.36–1.03)
Secondary/vocational	0.62 * (0.45–0.86)	0.61 * (0.39–0.95)	0.68 (0.420–1.09)
Tertiary	0.55 ** (0.39–0.77)	0.51 ** (0.327–0.80)	0.65 (0.402–1.07)
Employed	(1,1)	(1,1)	(1,1)
Unemployed	1.0 (0.88–1.14)	0.92 (0.77–1.10)	1.11 (0.91–1.34)
No comorbidity	(1,1)	(1,1)	(1,1)
HPT	1.26 ** (1.08–1.47)	1.26 * (1.02–1.54)	1.24 (0.98–1.56)
DM	1.37 * (0.99–1.88)	1.47 * (0.99–2.17)	1.24(0.72–2.16)
HPT/DM	1.82 ** (1.42–2.33)	1.67 ** (1.16–2.41)	1.88 ** (1.34–2.63)
Neurological diseases	2.03 (0.48–8.69)	1.88 (0.43–8.21)	0.59 (0.08–4.25)
Gastrointestinal diseases	2.08 * (1.25–3.46)	2.82 ** (1.41–5.61)	1.24 (0.58–2.66)
Pulmonary diseases	1.074 (0.73–1.59)	1.03 (0.62–1.71)	1.13 (0.60–2.11)
Cardiovascular diseases	1.344 (0.73–2.47)	1.22 (0.55–2.73)	1.48 (0.58–3.80)

Note: LOS—length of hospitalisation; HPT—hypertension; DM—diabetes mellitus; HPT/DM—hypertension and diabetes mellitus; * *p*-value < 0.05; ** *p*-value ≤ 0.01.

**Table 4 ijerph-19-00527-t004:** Determinants of long COVID (*N* = 2334). HPT/DM—hypertension and diabetes mellitus; * *p*-value < 0.05; ** *p*-value ≤ 0.01.

Variables	General Population (*N* = 2334) OR (95% CI)	Men (*N* = 1402) OR (95% CI)	Women OR (95%)
0–29 years	(1,1)	(1,1)	(1,1)
30–59 years	1.22 (0.47–3.20)	0.75 (0.24–2.41)	3.22 (0.54–19.07)
≥60 years	1.43 (0.42–4.83)	1.14 (0.26–4.92)	1.98 (0.18–22.1)
Men	(1,1)	-	-
Women	0.52 * (0.27–0.99)		
Ghanaian	(1,1)	(1,1)	(1,1)
Non-Ghanaian	0.48 (0.19–1.15)	0.59 (0.22–1.57)	0.25 (0.03–2.03)
No-formal	(1,1)	(1,1)	(1,1)
Primary	0.73 * (0.01–0.66)	-	0.59 (0.03–11.62)
Secondary/vocational	0.26 * (0.09–0.77)	0.23 * (0.06–0.83)	0.44 (0.04–5.24)
Tertiary	0.23 * (0.07–0.73)	0.17 * (0.04–0.69)	0.68 (0.06–8.21)
Employed	(1,1)	(1,1)	(1,1)
Unemployed	1.83 (0.35–1.95)	0.86 (0.31–2.39)	0.69 (0.13–3.59)
No comorbidity	(1,1)	(1,1)	(1,1)
Hypertension	1.23 (0.49–3.05)	1.29 (0.42–4.01)	1.34 (0.26–6.88)
HPT/DM	4.18 ** (1.61–10.85)	4.58 * (1.32–15.93)	5.69 * (1.08–30.16)

## Data Availability

All relevant data in this study are available at Ga East Municipal Hospital and can be accessed through the Ghana Health Service @ info@moh.gov.gh.

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
