# Peer review of "Determinants of COVID-19-Related Length of Hospital Stays and Long COVID in Ghana: A Cross-Sectional Analysis"

_ijerph, 2022, doi:10.3390/ijerph19010527_

Round 1
Reviewer 1 Report
Summary:
This manuscript analyzed the determinants for the longer length of COVID-19 related hospital stay and severity of COVID disease in Ghana. The authors found that patients with co-morbidities such as hypertension, diabetes, and gastrointestinal disorders spend a longer time in hospital and have high odds of severe COVID disease. Meanwhile, the education level has an opposite effect compared to co-morbidities. Since co-morbidities had a significant impact on the patient recovering from COVID disease, policies to address this healthcare crisis should prioritize the patients suffering from co-morbidities along with COVID.
Strengths:
1. The power of this study is that data was obtained from a large number of patients (n= 2334).
Weakness:
1. Even though the total number of patients enrolled in the study was high enough for statistical analysis, patients with prolonged COVID were minimal (2 %). Therefore, generalizing the study findings to a whole county with a 35 million population is not entirely accurate.
2. In order to analyze the correlation between independent variables and length of stay in the hospital, it is not clear the day cutoff used for separating patients with long COVID from without prolonged COVID.
3. The study is limited to a main COVID treatment center. Since the small hospitals were excluded, the variables that cause the prolonged hospital stay and severe COVID disease may differ, such as access to medical care and cultural and societal factors.
Minor comments:
1. In line 151, co-morbidities was misspelled as commodities.
2. In lines 16, 155, 162, there should be a space between a number and word days.
Author Response
Summary:
This manuscript analyzed the determinants for the longer length of COVID-19 related hospital stay and severity of COVID disease in Ghana. The authors found that patients with co-morbidities such as hypertension, diabetes, and gastrointestinal disorders spend a longer time in hospital and have high odds of severe COVID disease. Meanwhile, the education level has an opposite effect compared to co-morbidities. Since co-morbidities had a significant impact on the patient recovering from COVID disease, policies to address this healthcare crisis should prioritize the patients suffering from co-morbidities along with COVID.
Strengths:
- The power of this study is that data was obtained from a large number of patients (n= 2334).
Weakness:
- Even though the total number of patients enrolled in the study was high enough for statistical analysis, patients with prolonged COVID were minimal (2 %). Therefore, generalizing the study findings to a whole county with a 35 million population is not entirely accurate.
Response 1: Thank you for the comments. Yes, we agree that the findings of our study cannot be generalised to the wider Ghanaian population. We have, thus, included this limitation in our manuscript and this can be found in line 555 to 561 of the revised manuscript.
- In order to analyze the correlation between independent variables and length of stay in the hospital, it is not clear the day cutoff used for separating patients with long COVID from without prolonged COVID.
Response 2: The length of hospital stay was assessed as a count/continous variable. Therefore, we did not apply any cutoffs. For long COVID, we assessed it as patients who had COVID-19 symptoms lasting more than 4 weeks after the initial medical diagnosis. Consequently, the duration of symptoms differentiated long COVID from non-long COVID. See line 88 to 95 and line 155 to 162 of the manuscript for related information.
- The study is limited to a main COVID treatment center. Since the small hospitals were excluded, the variables that cause the prolonged hospital stay and severe COVID disease may differ, such as access to medical care and cultural and societal factors.
Response 3: Yes, we acknowledge the limitation associated with the use of a single treatment centre; thus, we have highlited the implication of this limitation on our findings in lines 555 to 561 of the revised manuscript.
Minor comments:
- In line 151, co-morbidities was misspelled as commodities.
Response 1: Please see the corrected spelling in line 216.
- In lines 16, 155, 162, there should be a space between a number and word days.
Response 2: Thank you for the comment. These corrections have been made accordingly and can be found in lines 21, 222 and 230 of the revised manuscript.
Reviewer 2 Report
1.In 2.2.2. Independent variables, comorbidities including pulmonary diseases, neoplasms, gastrointestinal diseases, cardiovascular diseases, and neurological diseases should be defined.
2.In line 151, commodities -> comorbidities
3.In line 165, "(14%)" is duplicated.
4.All abbreviations in tables should be denoted. In tables 3 and 4, denote Asterisk.
5.There are inconsistent alignment in tables 3 and 4.
6.In table 4, there is no data on men with primary education. Is there no patient in this subgroup ?
Author Response
Response to Comments from Reviewer 2
Comment 1: In 2.2.2. Independent variables, comorbidities including pulmonary diseases, neoplasms, gastrointestinal diseases, cardiovascular diseases, and neurological diseases should be defined.
Response 1: Many thanks for the comment. The comorbidities have been defined accordingly in line 179 to184 of the revised manuscript.
Comment 2: In line 151, commodities -> comorbidities
Response 2: Many thanks for pointing this out. This correction has been made and it can be found in line 216 of the edited manuscript.
Comment 3: In line 165, "(14%)" is duplicated.
Response 3: Thanks for the comment. The sentence containing the duplicate has been reviewed. See lines 232 to 239 of the revised manuscript.
Comment 4: All abbreviations in tables should be denoted. In tables 3 and 4, denote Asterisk.
Response 4: Thanks for the comments. A footnote describing the abbreviations and annotations have been added to the respective tables.
Comment 5: There are inconsistent alignment in tables 3 and 4.
Response 5: The alignment of tables 3 and 4 has been worked on.
Comment 6: In table 4, there is no data on men with primary education. Is there no patient in this subgroup ?
Response 6: Yes. There was no patient in this subgroup.
Reviewer 3 Report
I find the study reviewed interesting and innovative because it provides data on a subject that needs to be studied in greater depth in these times. Furthermore, it uses different independent variables that help to better explain a complex phenomenon such as COVID. Analysing this type of phenomenon can help in the prevention of risk factors and in the establishment of effective strategies, in this case, with respect to health services in Ghana.
It would be desirable not to repeat the same expressions in the title, keywords and abstract.
Overall, the structural guidelines of the journal are followed and it is understood that some existing formal issues will be addressed when the paper is finally accepted. The large sample of 2,334 patients treated is noteworthy. The results are adequately presented and the discussion is correct, highlighting the final recommendations to improve the existing situation in Ghana. In any case, the discussion could be extended, as the results provided are ample to be discussed, comparing them with others and suggesting explanations for the reasons behind the evidence.
Author Response
Response to comments from Reviewer 3
Comment 1: I find the study reviewed interesting and innovative because it provides data on a subject that needs to be studied in greater depth in these times. Furthermore, it uses different independent variables that help to better explain a complex phenomenon such as COVID. Analysing this type of phenomenon can help in the prevention of risk factors and in the establishment of effective strategies, in this case, with respect to health services in Ghana.
It would be desirable not to repeat the same expressions in the title, keywords and abstract.
Overall, the structural guidelines of the journal are followed and it is understood that some existing formal issues will be addressed when the paper is finally accepted. The large sample of 2,334 patients treated is noteworthy. The results are adequately presented and the discussion is correct, highlighting the final recommendations to improve the existing situation in Ghana. In any case, the discussion could be extended, as the results provided are ample to be discussed, comparing them with others and suggesting explanations for the reasons behind the evidence.
Response 1: Thank you very much for your comment. The abstract and keywords have been edited to incorporate relevant synonyms where appropriate (see line 10 to 30). The discussion section has also been revised accordingly (see line 460 to 492).
Reviewer 4 Report
Crankson et al.
for IJEnvResPubHealth
Determinants of COVID-19 related length of hospital stays and long COVID in Ghana: a cross-sectional analysis.
Summary:
The authors present a chart review of diagnosed COVID infections from the main COVID treatment centre in Ghana, a low-to-middle-income country with a national health system. They describe a striking increase in long COVID among people living with both diabetes mellitus and hypertension.
Comments:
Just a few revisions and an added figure are needed for clarity as described below.
Lines 39–44: medical costs will vary widely by locality due to resource availability and cost of delivery, and by country due to standard of care, legal, and economic pressures on the healthcare system. Furthermore, while COVID-19 is a global pandemic, this manuscript addresses paucity of data in Ghana. Please replace distractions from Columbia and Iran with available Ghanaian metrics, even if they require a bit of context. National Health System and proximity of care may be features of particular interest in the Ghanaian system, both for domestic policy and as a model for other countries. The proximity of care angle is generally difficult to get in chart reviews since blinding the data necessarily removes patient home addresses. The impact of distance on choosing to stay home or go to hospital is an interesting but difficult to access metric for researchers.
Table 1 and associated text, bivariate findings: Because the overwhelming majority of included patients did not have long COVID, the percentages overall are heavily skewed toward that population. The difference in the percentages for patient characteristics in the long COVID subset is limited in power by small sample size. These observations may be more informative if percentage definitions were approached in the orthogonal direction so that percentage adds to unity between yes and no long COVID. Thus 8/664 (1.2%), 30/1343 (2.2%), and 12/327 (3.7%) of patients developed long COVID by age. Men develop or are diagnosed with long COVID 5.1% of the time whereas women are diagnosed only 1.5% of the time (more than 3 times less, which is not accounted for by the disparity in presentation for care). Each of these variables predate SARS-CoV-2 infection. In the comorbidity context, 2.5% of people living with hypertension and 4.2% of people living with diabetes develop long COVID, but 8.8% of people living with both do so. 2.5% might not be statistically different from the 1.8% without comorbidity, and the sample set for diabetes is pretty small, but the dual comorbidity is striking. Also, the source of the length of stay p-value is unclear since it is measured as a length of stay in days.
Lines 193–197: Women are diagnosed with long COVID less often than men, but they are also turning up at the hospital less often too (not enough to account for the difference though). Some of this discrepancy could be due to behavioral factors like not complaining or physician bias, just as they appear less likely to turn up hospital for uncaptured reasons so "less likely to experience" is inaccurate, please change to less likely to be diagnosed or other correct phrase. Similar comment regarding "decreased odds of long COVID" on line 197.
Lines 237–248: This is a strawman argument and appears to be cherry-picked from high median age cohorts. The median age in the Makami paper was 59, Kaeuffer had a mean age of 66. John Ioannidis did a meta-analysis across studies in various countries (Bull World Health Organ 2021) and the results presented here are generally consistent with that report. Please revise. Also, please state the mean or median age of this cohort somewhere in the text.
Lines 249–261: The Ghanaian age pyramid is available data. Please include age pyramid graphs for COVID diagnosis, long COVID, and COVID-associated mortality in the cohort alongside the Ghanaian age pyramid. They would necessarily have to be scaled differently for legibility, but according to the statistics the comparison should be informative.
There are a number of minor errors in the text that the authors may wish to address. I have listed those I saw below, but there may be others. The authors should read through the entire text carefully or have a service do so.
Lines 16 and 161, 2days: 2 days.
Line 32, COVID-9: I believe the authors mean COVID-19.
Line 39 and 41, $: Please specify currency as USD or other with year as appropriate since such values change in meaning with inflation.
Line 41, UAS: Acronym lacks definition.
Line 166, associated with COVID-19 LOS: I believe the authors mean associated with increased COVID-19 LOS.
Line 182, 0-29years: 0–29 years
Tables 3 and 4 legends: describe * and ** annotations.
